

# Anti-COVID-19 multi-epitope vaccine designs employing global viral genome sequences

Tahreem Zaheer[1,*], Maaz Waseem[1,*], Walifa Waqar[1], Hamza Arshad Dar[1], Muhammad Shehroz[2], Kanwal Naz[1], Zaara Ishaq[1], Tahir Ahmad[1], Nimat Ullah[1], Syeda Marriam Bakhtiar[3], Syed Aun Muhammad[4] and Amjad Ali[1]

[1] Atta-ur-Rahman School of Applied Biosciences (ASAB), National University of Sciences and Technology (NUST), Islamabad, Pakistan
[2] Department of Biotechnology, Virtual University of Pakistan, Peshawar, Pakistan
[3] Department of Bioinformatics and Biosciences, Capital University of Science and Technology, Islamabad, Pakistan
[4] Institute of Molecular Biology and Biotechnology, Bahauddin Zakariya University, Multan, Pakistan
* These authors contributed equally to this work.

Corresponding author
Amjad Ali, amjaduni@gmail.com

## ABSTRACT

**Background:** The coronavirus SARS-CoV-2 is a member of the Coronaviridae family that has caused a global public health emergency. Currently, there is no approved treatment or vaccine available against it. The current study aimed to cover the diversity of SARS-CoV-2 strains reported from all over the world and to design a broad-spectrum multi-epitope vaccine using an immunoinformatics approach.

**Methods:** For this purpose, all available complete genomes were retrieved from GISAID and NGDC followed by genome multiple alignments to develop a global consensus sequence to compare with the reference genome. Fortunately, comparative genomics and phylogeny revealed a significantly high level of conservation between the viral strains. All the Open Reading Frames (ORFs) of the reference sequence NC_045512.2 were subjected to epitope mapping using CTLpred and HLApred, respectively. The predicted CTL epitopes were then screened for antigenicity, immunogenicity and strong binding affinity with HLA superfamily alleles. HTL predicted epitopes were screened for antigenicity, interferon induction potential, overlapping B cell epitopes and strong HLA DR binding potential. The shortlisted epitopes were arranged into two multi-epitope sequences, Cov-I-Vac and Cov-II-Vac, and molecular docking was performed with Toll-Like Receptor 8 (TLR8).

**Results:** The designed multi-epitopes were found to be antigenic and non-allergenic. Both multi-epitopes were stable and predicted to be soluble in an *Escherichia coli* expression system. The molecular docking with TLR8 also demonstrated that they have a strong binding affinity and immunogenic potential. These in silico analyses suggest that the proposed multi-epitope vaccine can effectively evoke an immune response.

## INTRODUCTION

COVID-19 has been declared a pandemic by the World Health Organization (WHO) as of the 11 March 2020 (*WHO, 2020*). The outbreak of SARS-CoV-2 infection starting in Wuhan, China has now affected millions of lives across the world. The common symptoms patients exhibit include fatigue, fever, dry cough, shortness of breath and upper airway congestion (*Chan, Wong & Tang, 2020*). The delay in symptom appearance due to the incubation period of the virus has caused more harm as infected people without symptoms might have already infected many others even before knowing they are carrying the infection. The number of infected people is increasing daily. Scientists are working tirelessly to find a cure, but as of now, the impasse continues.

Coronaviruses (CoVs) are a family of viruses causing diseases ranging from the common cold to more severe illnesses such as Severe Acute Respiratory Syndrome (SARS, caused by SARS-CoV) and Middle East Respiratory Syndrome (MERS, caused by MERS-CoV) (*Ceraolo & Giorgi, 2020b*; *Sahin, 2020*). Belonging to the family of Betacoronavirus, the latest CoV, recently named as SARS-CoV-2, is an enveloped spherical virion about 60–140 nm in diameter. It has a positive-sense single-stranded RNA genome of ~30 kb. The virus is believed to have originated from bats and is phylogenetically linked closest to the bat-SL-CoVZC45 and bat-SL-CoVZXC21 (*Zhu et al., 2020*). The virus is highly contagious and spreads from person-to-person through the respiratory droplets when an infected person sneezes or coughs or when a person encounters contaminated surfaces with the virus. The disease state caused by viral infection has been termed as COVID-19 (*Chan, Wong & Tang, 2020*).

The genome contains six open reading frames (ORFs), including ORF1ab, ORF3a, ORF6, ORF7a, ORF8, ORF10, encoding spike glycoprotein trimer S, nucleoprotein N, membrane protein M and envelopes small membrane protein pentamer E (*Zhu et al., 2020*). SARS-CoV-2 causes wide-ranging infections like mild upper respiratory tract disease, severe viral pneumonia with respiratory failure and severe cases, even death (*Huang et al., 2020*).

Considering the gravity of the situation of COVID-19 worldwide, there is a dire need for the development of an effective vaccine or antiviral drugs that are effective in treating this viral infection. Currently, there is no FDA approved treatment or licensed vaccine against COVID-19. Vaccination is the most desired option available at our disposal to eradicate the diseases associated with infectious microorganisms (*Greenwood, 2014*). Reverse vaccinology is a valuable technique that adopts genome-level analysis to identify potential antigenic determinants of the pathogenic microorganism (*Rappuoli, 2001*). Previously designed multi-epitope vaccines against various emerging viruses have been reported as safe and generate potent immune responses.

To elicit strong immunogenic responses and have long-term efficacy, vaccines against coronavirus must activate both humoral and cell-mediated immune responses (*Amanna & Slifka, 2011*). Virus-specific T cells, the effector T cells, provide vaccine-mediated protection at the peak of antiviral response and the antiviral cytokines production is also increased (*Sallusto, Geginat & Lanzavecchia, 2004*), but a successful vaccine would activate

B-cell mediated humoral immunity as well. In the case of re-infection, memory B cells can be re-activated via B cell receptors and provide protection (*Amanna, Carlson & Slifka, 2007*). Therefore, it is a challenge to activate both humoral as well as cell mediated immunity of required levels to effectively provide protection against COVID-19.

In this study, we have utilized the 475 genomes of SARS-CoV-2 available in NGDC (available at: https://bigd.big.ac.cn/) and GISAID (available at https://www.epicov.org/epi3/cfrontend) database as of 15 March 2020 to identify conserved immunogenic epitopes representing the novel coronavirus strains (collection) and to design a potent multi-epitope vaccine against the COVID-19 infection.

## MATERIALS AND METHODS

### Retrieval of sequences and multiple alignment

GISAID (available at https://www.epicov.org/epi3/cfrontend) and NGDC databases were mined to retrieve updated complete genomes of SARS-CoV-2. The retrieved genomes were then aligned using the Multiple Alignment using Fast Fourier Transform tool (MAFFT) (available at: https://www.ebi.ac.uk/Tools/msa/mafft/) (*Katoh et al., 2002*). Default parameters were used encompassing 200 PAM/$\kappa$ = 2 scoring matrix and a gap penalty of 1.53. These aligned sequences were visualized and analyzed in Unipro UGENE v.34 software (*Okonechnikov et al., 2012*) to generate a consensus sequence of all genomes and later, nBLAST was performed to analyze its diversity with Reference sequence NC_045512.2. The ORFs from this sequence were retrieved and subjected to further analysis of predicted proteins.

### CTL epitope mapping

CTLpred (available at: http://crdd.osdd.net/raghava/ctlpred/) was used for prediction of CD8 T cell epitopes (*Bhasin & Raghava, 2004*). Consensus approach was used that predicted CTL epitopes based on both Support Vector Machine (SVM) and Artificial Neural Network (ANN) methods using default thresholds (0.36 for SVM method and 0.51 for ANN method). For further analyses top 100 peptides were selected. These predicted epitopes were also verified by HLApred (available at: http://crdd.osdd.net/raghava/hlapred/) and ProPred1 (available at: http://crdd.osdd.net/raghava/propred1/) (*Singh & Raghava, 2003*).

### Antigenicity

All CTL epitopes were screened for antigenicity using VaxiJen (available at: http://www.ddg-pharmfac.net/vaxijen/) (*Doytchinova & Flower, 2007*). VaxiJen is an online server that predicts the antigenicity of peptides using the physicochemical properties of the proteins irrespective of their lengths and the need of alignments. We used VaxiJen server for our CTL epitopes using the default threshold 0.5 for viruses.

### Immunogenicity

After screening for the antigenic potential of CTL epitopes, epitopes having immunogenic potential were screened using the IEDB analysis resource for MHC class I Immunogenicity

(available at: http://tools.iedb.org/immunogenicity/) (*Calis et al., 2013*). This tool works best for 9-mer epitopes but larger lengths can be employed too. Only those epitopes that showed a strong binding affinity with HLA super family alleles were shortlisted. Default threshold was used for prediction of results. Epitopes having percentage rank less than 0.5 were qualified as strong binders.

## Strong binding affinity with HLA alleles and population coverage analysis

CTL Epitopes that were both antigenic and immunogenic were screened for their binding affinity with HLA superfamily alleles using Net MHC 4.0 (available at: http://www.cbs.dtu.dk/services/NetMHC/) (*Nielsen et al., 2003*; *Andreatta & Nielsen, 2016*). The tool predicts results on the basis of an ANN that has been trained using 81 different HLA alleles. MHC binding affinity of shortlisted epitopes was checked against HLA superfamily alleles (i.e., HLA-A0101, HLA-A0201, HLA-A0301, HLA-A2402, HLA-A2601, HLA-B0702, HLA-B0801, HLA-B2705, HLA-B3901, HLA-B4001, HLA-B5801, HLA-B1501). HLA supertype representative are clusters of HLA alleles based on their overlapping peptide binding specificity, hence can cover 80% of population in a less computationally exhaustive environment. The concept has been effectively utilized in prediction of promiscuously recognized epitopes against SARS, HCV, vaccinia virus and HBV (*Frahm et al., 2007*; *Sidney et al., 2008*). Furthermore, population coverage analysis was performed using IEDB resource server. The tool predicts population coverage of epitopes based upon the restriction database (*Bui et al., 2006*).

## HTL epitope mapping and population coverage analysis

The ORFs that didn't yielded prioritized CTL epitopes were subjected to Helper T Lymphocytes (HTL) epitope (CD4 T cell epitopes) prediction by the HLApred server. This was done to cover the whole proteome of the virus. All alleles of MHC-II were selected, and experimental as well as predicted binders were included in the analysis. A default threshold was used, and all predicted epitopes that shared no identity with humans were selected. The predicted epitopes were screened for their antigenic potential using VaxiJen as discussed earlier. The antigenic epitopes were further screened for their IFN-gamma induction potential using IFNepitope using hybrid approach that involves both SVM and Motif based approach. The prediction model chosen for analysis was IFN-gamma vs non IFN-gamma (available at: http://crdd.osdd.net/raghava/ifnepitope/scan.php) (*Dhanda, Vir & Raghava, 2013*). The epitopes screened positive for IFN-gamma were checked for overlapped B cell epitopes predicted by ABCpred (available at: http://crdd.osdd.net/raghava/abcpred/) using a default threshold (*Saha & Raghava, 2006b*). Lastly, the shortlisted epitopes were checked for their strong binding affinity with all the HLA DR alleles present in Net MHCII server (available at: http://www.cbs.dtu.dk/services/NetMHCII/). These alleles were DRB1_0101, DRB1_0103, DRB1_0301, DRB1_0401, DRB1_0402, DRB1_0403, DRB1_0404, DRB1_0405, DRB1_0701, DRB1_0801, DRB1_0802, DRB1_0901, DRB1_1001, DRB1_1101, DRB1_1201, DRB1_1301, DRB1_1302, DRB1_1501, DRB1_1602, DRB3_0101, DRB3_0202, DRB3_0301, DRB4_0101,

DRB4_0103, DRB5_0101. HLA-DR alleles are present on the surface of antigen-presenting cells and present the epitopes to helper T-cells initiating an immune response (*Smith-Garvin, Koretzky & Jordan, 2009*). HLA DRB alleles were reported to be involved in viral clearance, hence all the available HLA DRB alleles in server were selected for analysis (*Barrett, Ryan & Crowe, 1999*; *Keicho et al., 2009*). Lastly, Population coverage analysis was performed using IEDB population coverage analysis tool.

## Multi-epitope vaccine design and construction

Epitopes were arranged as per their arrangement in the reference sequence, the CTL epitopes were linked together using a flexible linker GGGGS as recommended by *Chen, Zaro & Shen (2013)*, and HTL epitopes with GPGPG linker to construct a multi-epitope (*Saadi, Karkhah & Nouri, 2017*). Moreover, a second multi-epitope construct was also designed where β-defensin, an adjuvant, was added at the N terminus of multi-epitope with EAAAK linker (*Shey et al., 2019*).

## Sequence-based physio-chemical properties of the multi-epitope design

Different physicochemical properties of multi-epitopes were predicted using ProtParam tool including prediction of in vivo half-life, theoretical isoelectric point (pI), molecular weight, instability index, aliphatic index and Grand Average of Hydropathy (GRAVY) index of peptides (*Wilkins et al., 1999*; *Gasteiger et al., 2005*). The degradation of peptides was based on amino acid at N-terminus hence N-end rule in Protparam was employed to predict in vivo half-life of both peptides. Instability index with a threshold of 40 was used in order to determine the stability of the protein in a cellular environment. The aliphatic and GRAVY index of both peptides was calculated based on their amino acid profiles.

## Allergenicity of multi-epitope

Both multi-epitope constructs (with and without adjuvant) were screened for their allergenicity using two online servers; AlgPred (available at: http://crdd.osdd.net/raghava/algpred/) (*Saha & Raghava, 2006a*) and Allergen FP (available at: http://ddg-pharmfac.net/AllergenFP/) (*Dimitrov et al., 2014*). In AlgPred, all four approaches were used including mapping of IgE epitopes and PID approach (allergenicity prediction is based on similarity with experimental IgE epitopes), BLAST search on allergen representative peptides (makes prediction by performing BLAST search against 2890 allergen representative peptides), MEME/MAST motif approach (predicts allergenic motifs using MAST) and hybrid approach. Allergen FP predicts the allergenicity of peptide using an alignment-free method by implementing a four-step algorithm and compared by Tanimoto coefficient.

## Antigenicity of multi-epitope design and modeling

The antigenicity profiles of both multi-epitope constructs were predicted again to confirm if the epitopes were antigenic or not. We used two servers this time; VaxiJen (available at: http://www.ddg-pharmfac.net/vaxijen/) (*Doytchinova & Flower, 2007*) and

AntigenPro (*Magnan et al., 2010*). Scratch Protein Predictor server available at http://scratch.proteomics.ics.uci.edu/ hosted the tool Antigen Pro for antigenicity estimation. AntigenPro, like VaxiJen, predicts antigenicity using physicochemical properties, irrespective of the length of peptides as well as its alignment.

## Secondary structure prediction

For evaluation of further structural characteristics of designed multi-epitopes, PDBsum (available at http://www.ebi.ac.uk/thornton-srv/databases/pdbsum/) (*Laskowski et al., 2018*) was used. PDBsum provides insight into the unique structural aspects of proteins, peptides and their ligands.

## Tertiary structure prediction and refinement

Tertiary structures of both vaccines were predicted by Scratch Protein Predictor-hosted 3Dpro tool (available at http://scratch.proteomics.ics.uci.edu/) (*Cheng et al., 2005*) and refined by Galaxy refine (available at http://galaxy.seoklab.org/cgi-bin/submit.cgi?type=REFINE) (*Ko et al., 2012*). The model showing highest Ramachandran favored residues and minimum poor rotamers were selected. The refined structures were checked for their stability and flexibility using molecular dynamics (MD) simulation studies. Structural flexibility of a protein/peptide is important for its molecular recognition and its function, so a coarse-grained protein model implemented in a webserver CABS-Flex 2.0 (available at: http://biocomp.chem.uw.edu.pl/CABSflex/) (*Kuriata et al., 2018*) was used for near-native dynamics of both vaccines. Default distance restrained parameters were used, number of cycles and cycles between trajectory frames were raised to 100. Temperature of simulation was also kept default.

## Molecular docking with Toll-Like Receptor 8

The models were then checked for their interaction with Toll-Like Receptor 8 (TLR8). Active and passive residues for these analyses were predicted using CPORT (*De Vries & Bonvin, 2011*). Molecular docking was performed by using guru level interface of HADDOCK 2.2 (available at: http://haddock.science.uu.nl/services/HADDOCK/haddockserver-guru.html) (*Van Zundert et al., 2016*) with default parameters and a representative structure from the top-ranked docked cluster with minimum HADDOCK score was refined using Refinement Interface (available at: http://haddock.science.uu.nl/services/HADDOCK/haddockserver-refinement.html) (*Van Zundert et al., 2016*). Moreover, NMA analysis of refined complexes was also performed using iMOD in order to determine their deformation potential (*Kremer, Mastronarde & McIntosh, 1996*) (available at: https://bio3d.colorado.edu/imod/paper/). The interacting residues between each complex were predicted using PDBsum analysis (*Laskowski et al., 2018*).

## Codon optimization and in silico cloning in *Escherichia coli*

Codon optimization of both multi-epitopes was performed using JCAT (available at: http://www.jcat.de/) (*Grote et al., 2005*) and codons were adapted as per *E. coli* K12 strain.

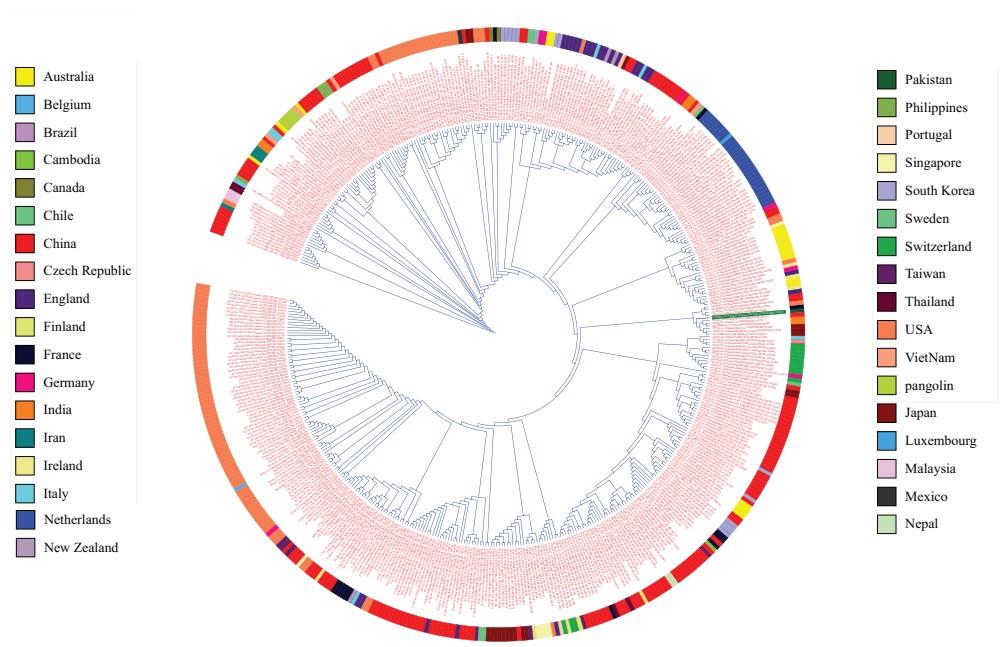

| | |
|---|---|
| Australia | Pakistan |
| Belgium | Philippines |
| Brazil | Portugal |
| Cambodia | Singapore |
| Canada | South Korea |
| Chile | Sweden |
| China | Switzerland |
| Czech Republic | Taiwan |
| England | Thailand |
| Finland | USA |
| France | VietNam |
| Germany | pangolin |
| India | Japan |
| Iran | Luxembourg |
| Ireland | Malaysia |
| Italy | Mexico |
| Netherlands | Nepal |
| New Zealand | |

**Figure 1 Phylogenetic tree analysis of 475 complete genomes of SARS-CoV-2 based on full genome nucleotide sequences using the UPGMA.** Outer ring displays genomes with different colors for each country.

Default parameters were used and plasmid UC19 was used for in silico cloning of both multi-epitopes. Moreover, a His6 tag was added at the both ends of sequences for the purification of multi-epitope proteins.

## RESULTS

### Retrieval of sequences and multiple sequence alignments

The recruited 475 complete SARS-CoV-2 genomes showed a high level of conservancy upon multiple sequence alignments and in the phylogenetic tree (Fig. 1). The consensus sequence was found to be 99% identical, with 10 gaps and no mismatches with Ref seq NC_045512.2 from Wuhan. There were 11 ORFs available in Refseq NC_045512.2 for protein coding sequence (ORF1ab (YP_009724389.1), Spike Glycoprotein (YP_009724390.1), ORF3a (YP_009724391.1), Envelope Protein (YP_009724392.1), Membrane Glycoprotein (YP_009724393.1), ORF6 (YP_009724393.1), ORF7a (YP_009724393.1), ORF7b (YP_009725318.1), ORF8 (YP_009724396.1), ORF9 (YP_009724397.2) and ORF10 (YP_009725255.1)). These ORFs were subjected to epitope mapping.

### CTL epitope mapping

A total of 13 CTL epitopes were shortlisted for designing multi-epitopes. These shortlisted epitopes were predicted to be antigenic (as determined by VaxiJen tool), immunogenic (as determined by MHC-I IEDB Immunogenicity tool), and had potential to bind with HLA superfamily alleles (as predicted by NetMHC) (Table 1). All the screened epitopes

**Table 1 Shortlisted CTL epitopes for multi-epitope design.**

| Sr. No. | Position | Epitope | CTLpred score (SVM/ANN) | MHC | Antigenicity | Immunogenicity | Strong Binding |
|---|---|---|---|---|---|---|---|
| ORF1ab | | | | | | | |
| 1 | 1561 | SLREVRTIK | 0.97/1.3171709 | 15 | 0.7073 | 0.31699 | 67.66 0.40 <= SB HLA-A0301 |
| 2 | 2363 | WLMWLIINL | 1.00/1.1904209 | 10 | 1.0138 | 0.38891 | 6.60 0.06 <= SB HLA-A0201 |
| 3 | 214 | TLSEQLDFI | 0.90/0.83436257 | 11 | 1.0499 | 0.01093 | 26.42 0.40 <= SB HLA-A0201 |
| 4 | 5679 | YVFCTVNAL | 0.77/0.92267704 | 10 | 0.5377 | 0.07781 | 10.00 0.03 <= SB HLA-B3901 |
| 5 | 2351 | FSYFAVHFI | 0.66/0.97709163 | 13 | 0.8806 | 0.28926 | 133.13 0.50 <= SB HLA-B5801 |
| 6 | 2335 | TRFFYVLGL | 0.87/0.7530986 | 10 | 0.5606 | 0.2015 | 40.93 0.17 <= SB HLA-B2705 |
| 7 | 5255 | AIDAYPLTK | 0.82/0.78760102 | 10 | 0.5972 | 0.04585 | 95.81 0.50 <= SB HLA-A0301 |
| Surface glycoprotein | | | | | | | |
| 8 | 350 | VYAWNRKRI | 0.93/0.49736819 | 11 | 0.5003 | 0.12625 | 0.30 <= SB HLA-A2402 |
| 9 | 1060 | VVFLHVTYV | 0.59/0.77948535 | 10 | 1.5122 | 0.1278 | 0.50 <= SB HLA-A0201 |
| Membrane glycoprotein | | | | | | | |
| 10 | 136 | SELVIGAVI | 0.92/0.48139255 | 13 | 0.6409 | 0.25658 | 0.15 <= SB HLA-B4001 |
| ORF6 | | | | | | | |
| 11 | 3 | HLVDFQVTI | 0.59/1.0193905 | 13 | 1.4119 | 0.0982 | 0.40 <= SB HLA-A0201 |
| ORF7a | | | | | | | |
| 12 | 101 | FLIVAAIVF | 0.53/0.37735924 | 9 | 0.586 | 0.29611 | 0.25 <= SB HLA-B1501 |
| ORF7b | | | | | | | |
| 13 | 26 | IIFWFSLEL | 0.88/0.96108642 | 9 | 0.8291 | 0.2683 | 0.40 <= SB HLA-A0201 |

Note:
  CTL epitopes presenting antigenicity, immunogenicity and strong immunogenic potential shortlisted for designing multi-epitopes.

had 88.42% population coverage. The details of epitopes predicted from each ORFs filtered at each stage along with their properties have been provided in Table S1.

## HTL epitope mapping

A total of 10 HTL epitopes were scrutinized for multi-epitopes. These shortlisted epitopes were predicted to lack homology with Human proteome (as determined by HLApred), had overlapping B cell epitopes (as determined by ABCpred), showed potential to induce IFN-gamma response (as determined by IFNepitope server) and demonstrated strong binding affinity with HLA DR alleles (as predicted by NetMHCII) (Table 2). All the epitopes had 89.70% population coverage (as predicted by IEDB population coverage analysis tool). The detail of step wise scrutiny and all the initially predicted epitopes is provided in Table S2.

## Multi-epitope structure design and modeling

Two multi-epitope constructs were designed with and without β-defensin adjuvant to analyze and compare whether the designed multi-epitope is efficacious only in presence of adjuvant or has the ability to elicit immune response solely. Cov-I-Vac was the multi-epitope construct without adjuvant and Cov-II-Vac was with adjuvant (Fig. 2).

**Table 2 HTL epitopes prioritized for multi-epitope vaccine design.**

| Sr. No. | Epitope | Antigenic | IFN | Binding affinity | Affinity (nM) | MHC |
|---|---|---|---|---|---|---|
| Envelope | | | | | | |
| 1 | FLLVTLAIL | 0.9645 | 0.103076 | DRB1_0103, DRB3_0301 | 9,440.3, 654.2 | 11 |
| ORF8 | | | | | | |
| 2 | LGIITTVAA | 0.8663 | 0.10392924 | DRB1_0801 | 646.6 | 30 |
| 3 | WYIRVGARK | 1.1953 | 0.10404837 | DRB5_0101, DRB4_0103, DRB1_0801 | 661.7, 25.5, 228.8 | 10 |
| ORF9 | | | | | | |
| 4 | IGYYRRATR | 0.888 | 0.47631827 | DRB4_0103, DRB1_0801 | 62.7, 480.3 | 12 |
| 5 | VILLNKHID | 0.5902 | 0.18486858 | DRB1_0801 | 309.6 | 22 |
| ORF3a | | | | | | |
| 6 | IITLKKRWQ | 1.7866 | 0.16918449 | DRB1_0801 | 668.3 | 22 |
| 7 | ITLKKRWQL | 1.9347 | 0.41796429 | DRB1_1301, DRB4_0103 | 32.8, 191.2 | 14 |
| 8 | LLLVAAGLE | 0.8981 | 0.03978107 | DRB1_0801 | 763 | 46 |
| 9 | VRATATIPI | 0.7515 | 0.18967277 | DRB1_0701, DRB1_1302, DRB3_0301 | 13.2, 527.3, 120.8 | 25 |
| 10 | FLCWHTNCY | 0.8927 | 0.26613992 | DRB1_0402 | 8,117.1 | 9 |

**Note:**

CD4 T cell epitopes having strong antigenicity, overlapping B cell epitopes and IFN induction potential that were screened to design multi epitopes.

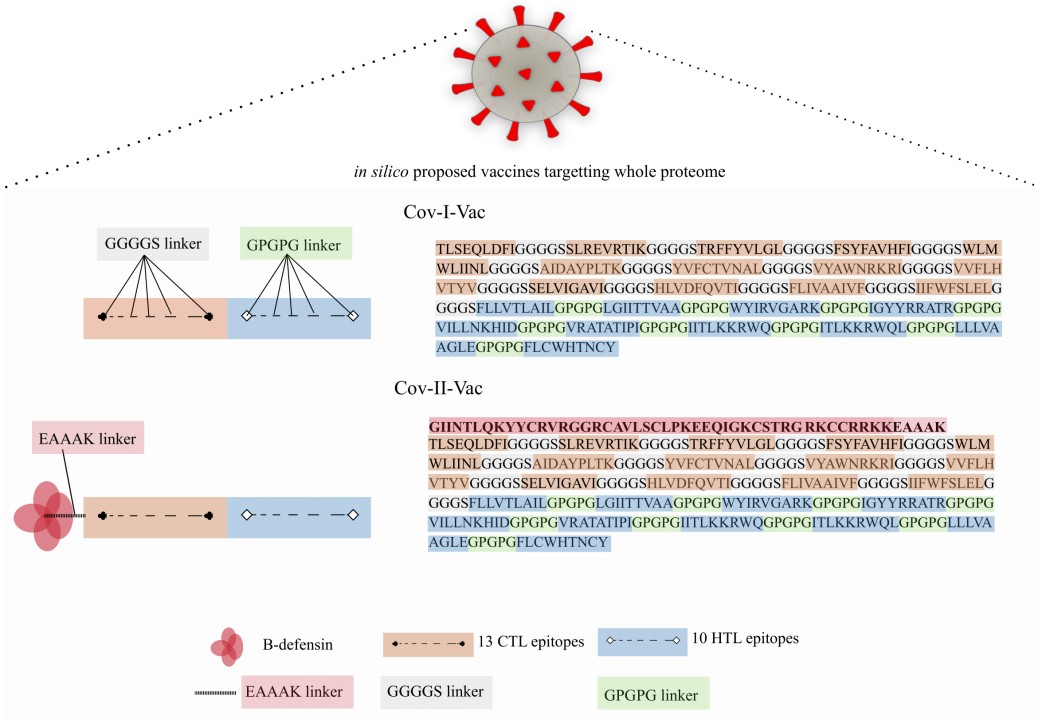

**Figure 2 Graphical representation of Cov-I-Vac and Cov-II-Vac.** All the available ORFs were subjected to epitope mapping and shortlisted epitopes were used to design Cov-I-Vac. Furthermore, beta defensin adjuvant was added at N-terminus to design Cov-II-Vac. Cov-I-vac and Cov-II-vac have CTL as well as HTL epitopes to incite cell mediated as well as humoral immunity.

## Sequence-based physio-chemical properties of multi-epitope

Physio-chemical analysis of Cov-I-Vac and Cov-II-Vac revealed that both constructs are reasonably stable (38.48 and 39.91 respectively), have molecular weight <50 kDa, lower GRAVY indexes (i.e., 0.394 and 0.249, respectively), with a pI of 9.97 and 10.03, respectively. The N terminus of Cov-I-Vac has Threonine so its half-life in mammalian reticulocytes was predicted to be 7.5 h (if tested in vitro) using Protparam. Meanwhile, Cov-II-Vac, because of Glycine at N terminus, has a half-life of >30 h in mammalian reticulocytes, as predicted using N-end rule. However, no difference in in vivo half-life was predicted, and was found to be >20 h in yeast and >10 h in *E. coli*.

## Antigenicity, allergenicity and immunogenicity of Cov-I-Vac and Cov-II-Vac

The antigenicity potential was found to be retained in both multi-epitopes and was estimated as 0.6153 and 0.5947 for Cov-I-Vac and Cov-II-Vac, respectively. Both constructs were reported as antigenic by AntigenPro as well with an antigenicity score of 0.516406 and 0.386605.

Moreover, both vaccines were soluble upon overexpression in *E. coli*. The safety profile also showed that constructs do not exhibit any experimentally reported allergens and thus projected to be non-allergens.

## Secondary structure of Cov-I-Vac and Cov-II-Vac

The secondary structure prediction of Cov-I-Vac shows that it has a sheet and beta hairpin between two strands (Ala160-Il161, Gly164-Gly165), six helices, 42 beta turns and two gamma turns (Fig. 3A). The secondary structure of Cov-II-Vac shows that it has a sheet and two beta hairpins among three strands (Val193-Thr198, Gly201-Val208, Gly216-Gly217), 10 helices, five helix-helix interactions, and 37 beta turns (Fig. 3B).

## Tertiary structure prediction and refinement

The tertiary structure of Cov-I-Vac had 10.2% poor rotamers with 90.5% Rama favored residues; however, the refined structure selected for further analysis had an RMSD of 0.541 with 0% poor rotamers and 93% Rama favored residues. Similarly, the initial structure of Cov-II-Vac had 5.9% poor rotamers with 89% Rama favored residues, but the refined shortlisted structure with RMSD 0.548 had 92.9% Rama favored residues and 0% poor rotamers.

## CABS-flex analysis

The trajectory of 10 models was generated after simulation (Fig. 4). The Root Mean Square Fluctuation (RMSF) of this simulation was found to be between 3.9 and 5.8 Å.

## The trajectory of Cov-II-Vac after fast simulation

Trajectory analysis of fast simulation of Cov-II-Vac was analyzed and the RMSF of the structure was found to be 0.75–7.25 Å. The maximum fluctuation was observed at residue 227 while minimum fluctuation was residue 39–59 (Fig. 5).

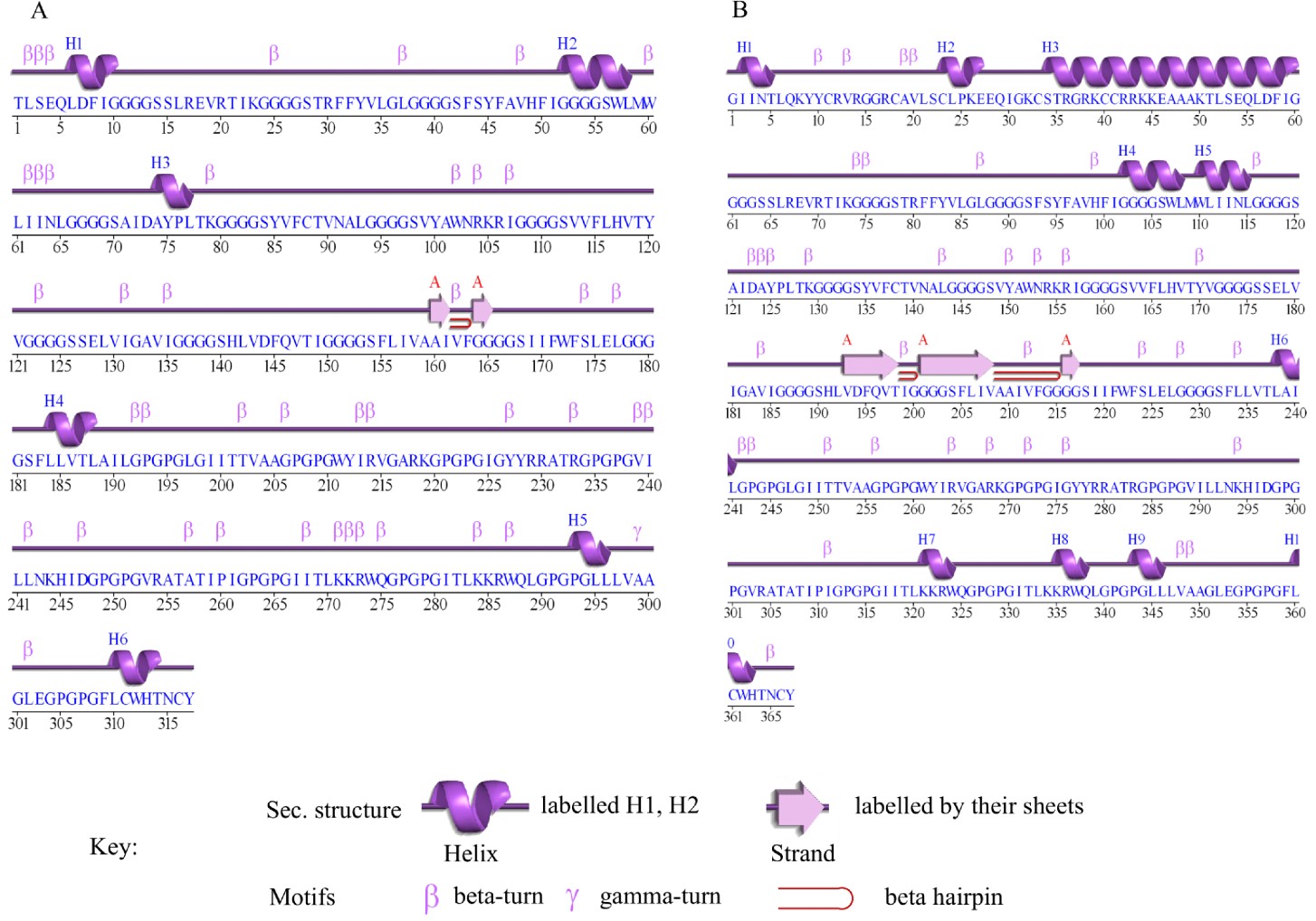

**Figure 3 Secondary structure of multi-epitope vaccines designed against SARS-CoV-2.** (A) Secondary structure of Cov-I-Vac. (B) Secondary structure of Cov-II-Vac.

The active and passive residues of Cov-I-Vac, Cov-II-Vac and TLR8 are provided in Table S3. The molecular docking of Cov-I-Vac with TLR8 resulted in the top-ranked cluster with a HADDOCK score of −40.8 +/− 5.5. Low values of HADDOCK score, especially those that are negative, usually imply significantly high interaction between proteins. HADDOCK refined the representative structure of this complex and clustered the resulting 20 structures into one complex that represented 100% of the refined water models generated. After undergoing molecular refinements, the HADDOCK score of the Cov-I-Vac-TLR8 complex improved significantly to reach −251.3 +/− 2.3. The details of the refined interaction along with statistical parameters are provided in Table 3. Cov-II-Vac also showed significant interaction with TLR8 that was −83.5 +/− 6.3 which was a higher interaction score than Cov-I-Vac. However, after refinements, the interaction of Cov-II-Vac with TLR8 was lower compared to Cov-I-Vac-TLR8 interaction, that is, −217.4 +/− 3.5. The details of HADDOCK score with all its parameters are mentioned in Table 3.

A

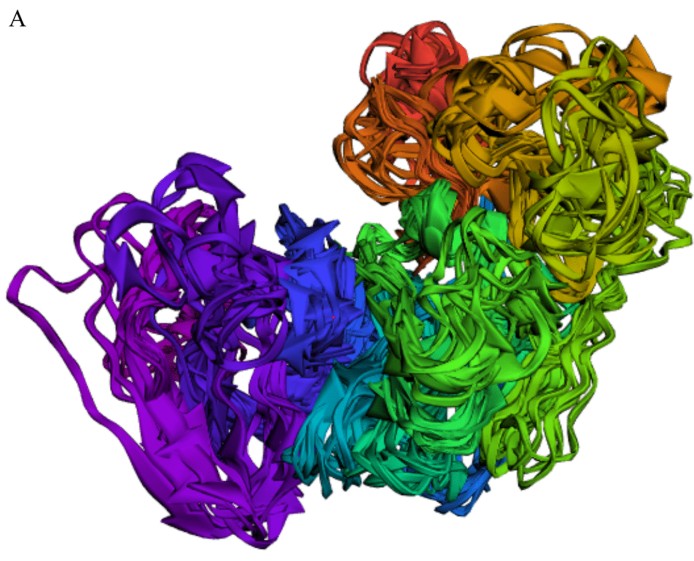

B

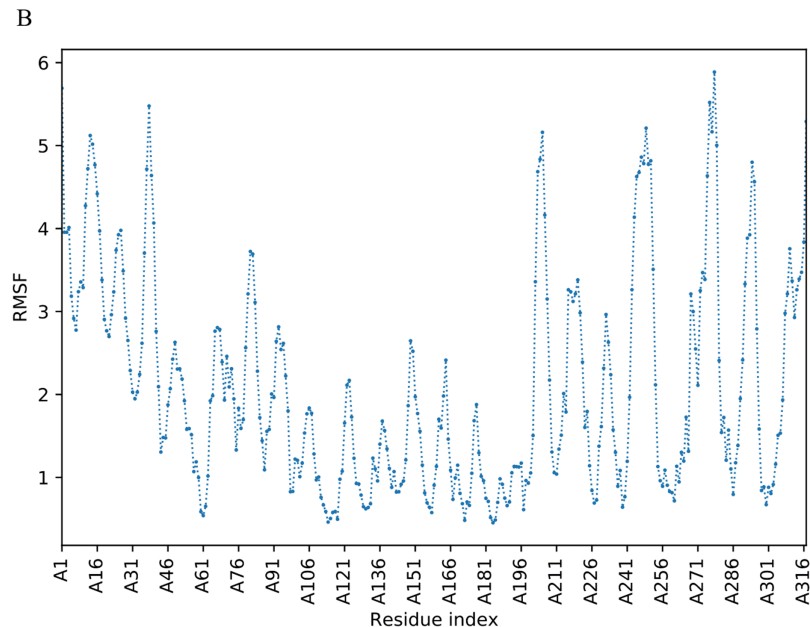

**Figure 4 Fast dynamic simulation of Cov-I-Vac.** (A) Trajectory analysis of Cov-I-Vac showing that multi-epitope remained fairly stable during the perturbations. (B) RMSF of multi-epitope during simulation.                                                                           

For an in depth analysis of protein–protein interactions, PDBSum analysis revealed that a total of 36 residues of Cov-I-Vac interact with 43 residues of TLR8, with one salt bridge, 15 hydrogen bonds and 210 non bonded interactions. The detail of the interaction is illustrated in Fig. 6 and atomic level interaction is provided in Table S4. Similarly, 36 residues of Cov-II-Vac interacted with 41 residues of TLR8, prominent among them were residues of multi-epitopes of 12 hydrogen bonds and 168 non bonded contacts. The interacting residues of the complex are illustrated in Fig. 7 while distances of detailed atomic interaction are provided in Table S5.

A

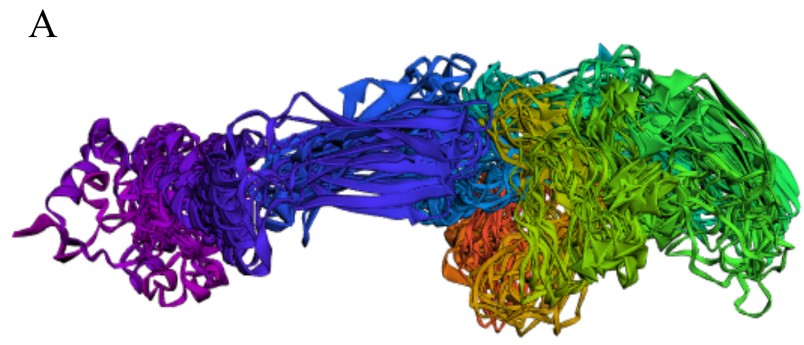

B

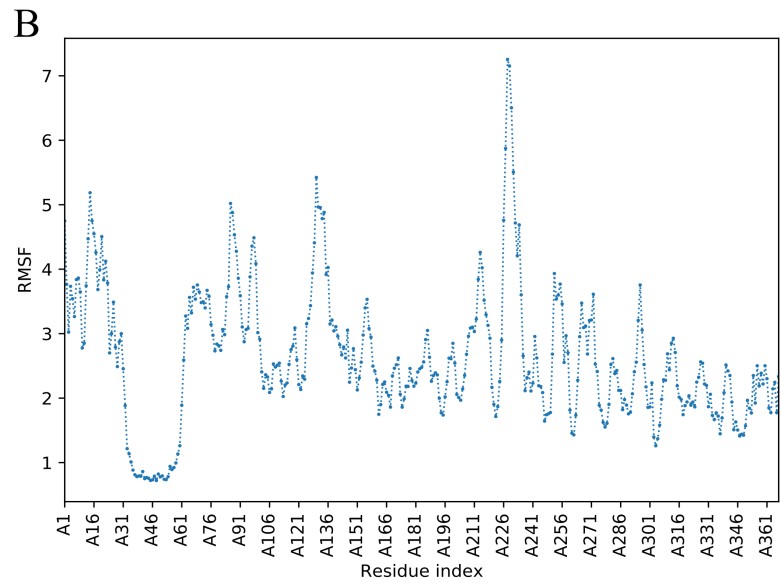

**Figure 5 Trajectory analysis of Cov-II-Vac using CABS-flex analysis.** (A) Trajectory of multi-epitope protein (B) RMSF of multi-epitope during simulation.   

**Table 3 Molecular docking score of multi-epitopes with TLR8.**

| Parameters | Cov-I-Vac | Cov-II-Vac |
|---|---|---|
| HADDOCK score | −251.3 ± 2.3 | −217.4 ± 3.5 |
| Cluster size | 20 | 20 |
| RMSD from the overall lowest-energy structure | 0.3 ± 0.2 | 0.3 ± 0.2 |
| Van der Waals energy | −165.3 ± 6.6 | −149.7 ± 2.7 |
| Electrostatic energy | −305.3 ± 21.6 | −160.7 ± 21.6 |
| Desolvation energy | −25.0 ± 5.4 | −35.6 ± 3.9 |
| Restraints violation energy | 0.9 ± 0.09 | 0.6 ± 0.25 |
| Buried Surface Area | 4,417.1 ± 74.1 | 4,225.6 ± 40.0 |
| Z-Score | 0 | 0 |

**Note:**
  Both Cov-I-Vac and Cov-II-Vac show a strong binding affinity with Immune receptopr TLR8.

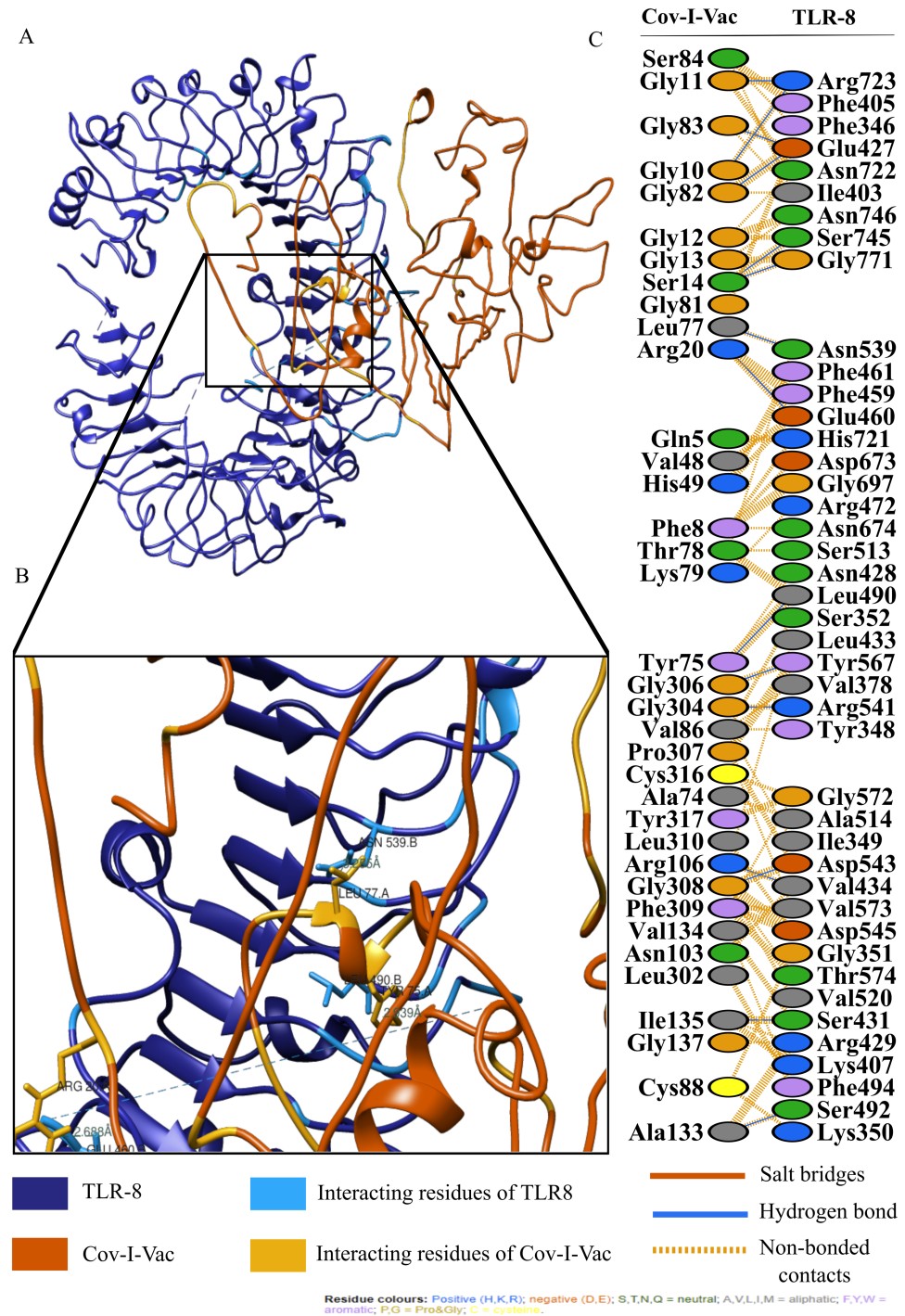

**Figure 6 Protein-protein interaction of Cov-I-Vac and TLR8.** (A) Refined complex of Cov-I-Vac interacting with TLR8. (B) Atomic level interaction showing Hydrogen bond and non-bonded contacts between residues of Cov-I-Vac and TLR8. (C) List of all the residues of both interacting proteins.

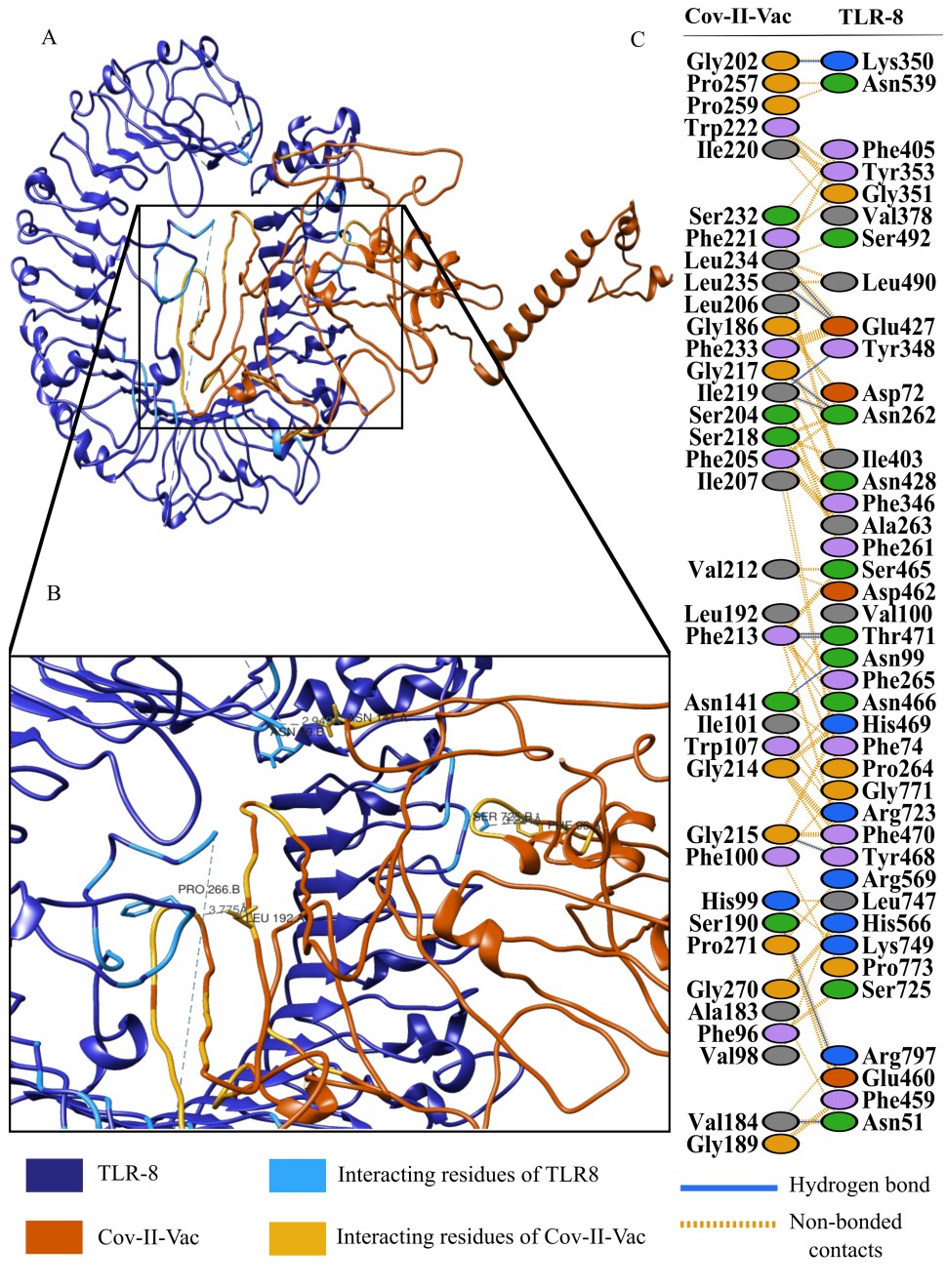

**Figure 7 Protein-protein interaction of Cov-II-Vac and TLR8.** (A) Refined complex of Cov-II-Vac interacting with TLR8. (B) Atomic level interaction showing Hydrogen bond and non-bonded contacts between residues of Cov-II-Vac and TLR8. (C) List of all the residues of both interacting proteins.

## NMA analysis

NMA analysis of both multi-epitopes showed that Cov-I-Vac has higher stability than Cov-II-Vac, as the Eigenvalue, the energy required to deform the structure, was higher for Cov-I-Vas as shown in Fig. 8.

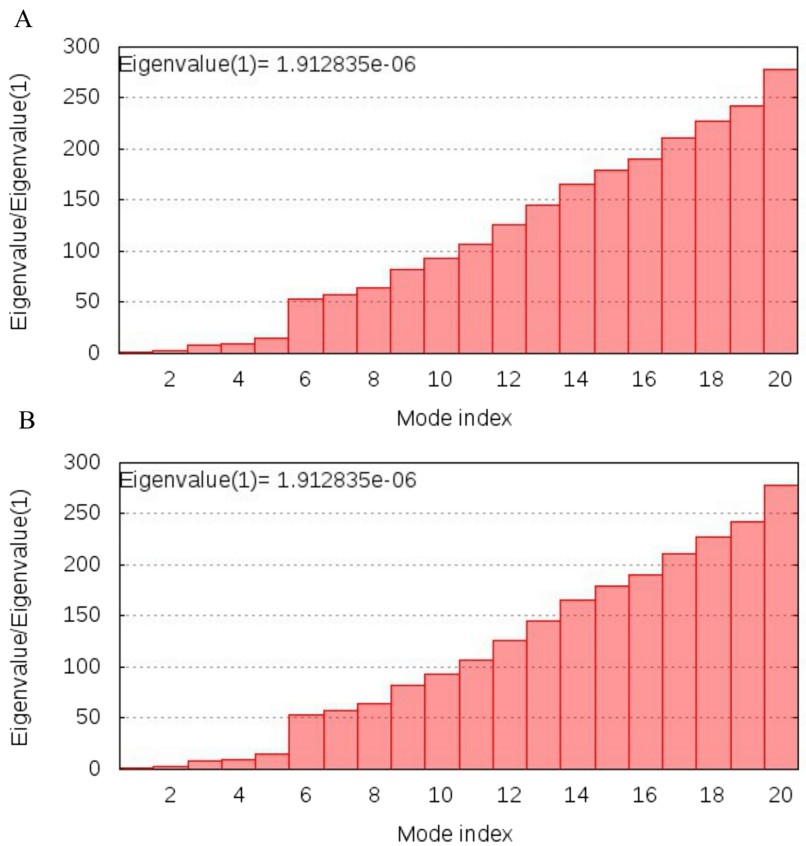

**Figure 8 Normal Mode analysis of designed multi epitopes.** (A) Eigen score of Cov-I-Vac and TLR8 complex. (B) Eigen score of Cov-II-Vac and TLR-8 complex.

## Codon optimization and *E. coli* expression

Codon optimization results of both vaccines Cov-I-Vac and Cov-II-Vac (with GC content 58.99% and 58.49%, respectively) were within the optimum limits (*Pandey, Bhatt & Prajapati, 2018*). The CAI value was predicted to be 1.0, which projects high level expression of our designed vaccine constructs in *E. coli* K12 strain. The total length of the Cov-I-Vac clone was 3.6 kbp while Cov- II-Vac clone was 3.8 kbp. Both sequences were designed to be added to the plasmid between restriction sites AflIII-pciI and BspQI-sapI (Fig. 9).

## DISCUSSION

The COVID-19 pandemic has affected 213 countries around the world with almost 6.3 million patients and more than 378 thousand deaths (*Worldometersinfo, 2020*). The current situation urges scientists all over the world to find an urgent solution to stop this pandemic and develop effective therapeutics (*WHO, 2020*). To ensure viral clearance, cell mediated and humoral responses must be induced by the action of CD8 and CD4 T cells (*Ikram et al., 2018*). The reliability of such responses has enabled the use immunoinformatics approaches for vaccine development against viral diseases. The present study encompassing 475 complete genomes from all around the world

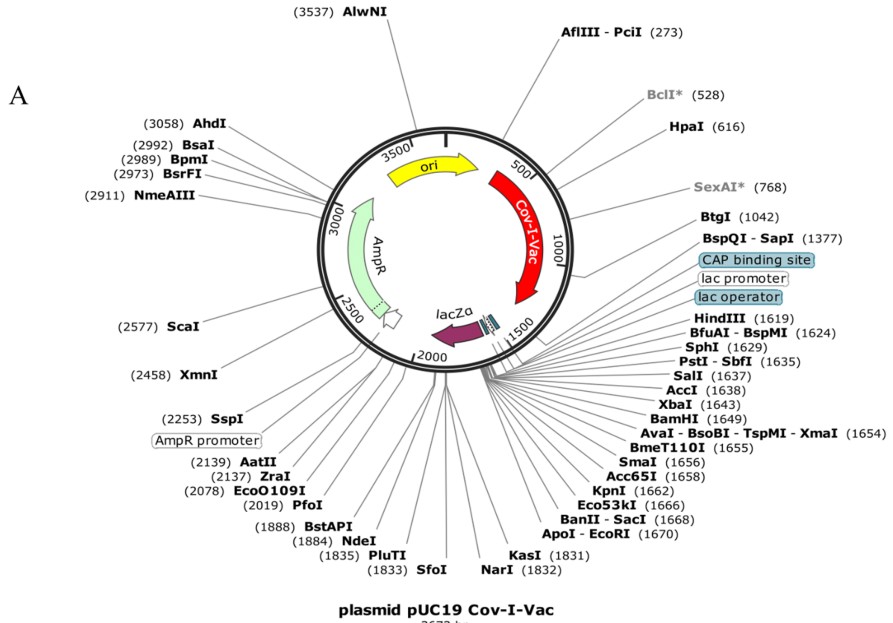

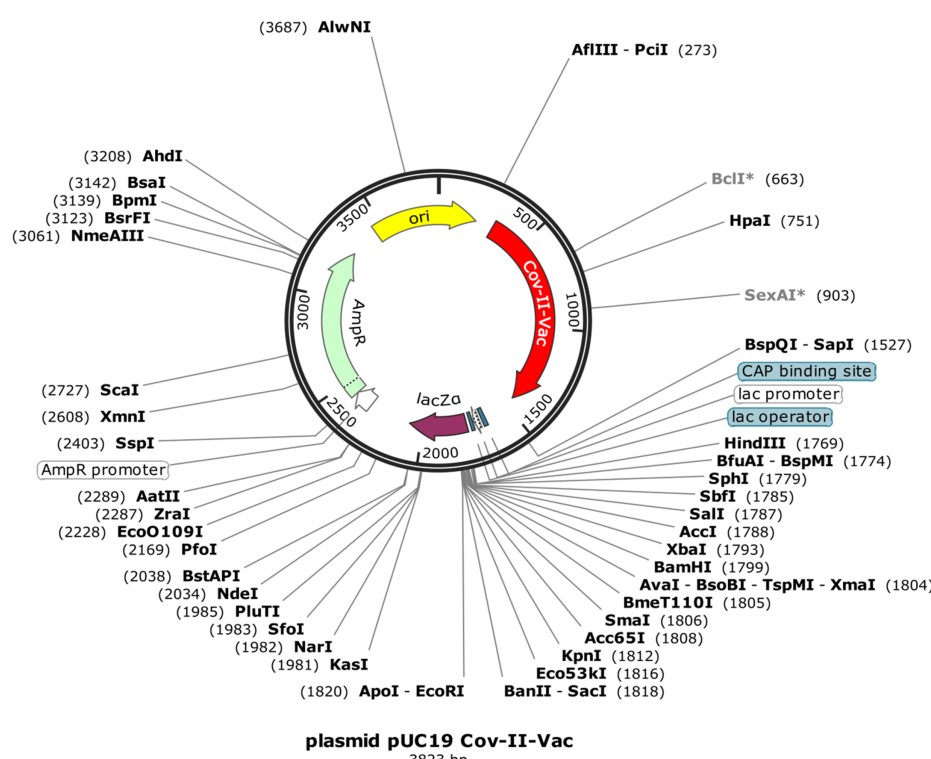

**Figure 9 In silico cloned multi epitopes in plasmid pUC19.** The CAI values indicate that both multi-epitopes have high expression in *E. coli* expression system. (A) Cov-I- vac cloned in pUC19. (B) Cov-II-Vac cloned in pUC19.           

suggests that while there is emerging variation in SARS-CoV-2, the rate with which this variation arises is quite slow (*Ceraolo & Giorgi, 2020a*). The consensus sequence generated from the alignment of all complete genomes was found to be 99% identical to the reference sequence reported from Wuhan.

The phylogenetic analysis of 475 genomes of SARS-CoV-2 reveals that genome sequences reported from China belong to clades containing those from almost all of the countries of the world and covering the whole span of a phylogenetic tree. The relatedness of SARS-Cov-2 strains was expected considering that the outbreak originated from China (*Nezafat et al., 2016*; *Mackenzie & Smith, 2020*; *Wu et al., 2020a*).

The present study was mainly conducted to propose multi-epitope based vaccines against SARS-CoV-2 by an immunoinformatics approach similar to other studies (*Tomar & De, 2010*; *Dar et al., 2016*; *Ikram et al., 2018*). These vaccines are advantageous compared to other monovalent vaccines because they provide superior protection by eliciting strong humoral and cell mediated immunity (*Amanna & Slifka, 2011*). Three subunit vaccines have been previously reported that were based on one, three, and 10 proteins (*Abdelmageed et al., 2020*; *ul Qamar et al., 2020*).

In recent years, epitope-based vaccines, developed via immunoinformatics approach, have gained global recognition as an alternate therapeutic strategy because of the various advantages they possess. A multi-epitope vaccine was designed against MERS, a virus belonging to the same family of viruses as SARS-CoV-2, using in silico tools (*Srivastava et al., 2018*). Another multi-epitope peptide vaccine was designed against brucellosis which elicited strong T cell responses (*Saadi, Karkhah & Nouri, 2017*). *Shey et al. (2019)* reported a similar approach used for prophylactic/therapeutic multi-epitope vaccine design for Onchocerciasis caused by microflaria. Promising results were shown in mice models by a multi-epitope vaccine developed by *Lin et al. (2016)* against Epstein-Barr virus, designed using bioinformatics tools. These studies highlight the importance of multi-epitope vaccine design to combat infections, and validate the methodology adopted in our study.

The vaccine designed in our study used all predicted ORFs and therefore covers the complete proteome. Epitopes selected were 13 CTL based and 10 HTL based that elicit cell mediated and humoral immune responses, respectively. Therefore, our subunit vaccines are projected to be effective and immunogenic against SARS-Cov-2. Coding sequences of the proteins of SARS-CoV-2 from NCBI were evaluated for their antigenicity. Highly antigenic sequences were used to predict B and T cell epitopes, the key step in vaccine development. Antigenic T cell epitopes were screened to overlap with B-cell and IFN-gamma epitopes (*Van Regenmortel, 1996*). HTL epitopes overlapping with B cell epitopes were further proceeded for multi-epitope construction. HTL epitopes overlapped with B-cell epitopes have been used in vaccine design studies to generate humoral response (*Lehtinen et al., 1995*; *Dar et al., 2019*). Cellular immunity mediated by CTL and humoral immunity mediated by HTL epitopes can aid viral clearance from body. Subunit vaccines proposed against HIV using HTL and CTL epitopes are under clinical trials (*Newman et al., 2002*).

The multi epitope vaccines, proposed in this study, were designed with the help of linkers; epitopes were linked with GGGGS (CTL based) and GPGPG (HTL based) linkers. Linkers were added to help maintain the proper functioning of each epitope after being imported to the human body independently (*Pandey, Bhatt & Prajapati, 2018*). GGGGS linker was added between CTL epitopes to ensure flexibility as well as solubility and pH stability (*Chen et al., 2017*). Addition of the GPGPG linker induces strong HTL responses as it not only facilitates the processing and presentation of epitopes but also prevents the formation of junctional epitopes in order to maintain immunogenicity (*Ikram et al., 2018*; *Wu et al., 2020b*).

An adjuvant, β-Defensin was added to the N-terminus of Cov-I-Vac to make it Cov-II-Vac, using EAAAK linker, in order to increase the immunogenicity of the vaccine. N-terminus amino acid plays a role in ubiquitin mediated degradation (*Wilkins et al., 1999*). The EAAAK linker effectively separates the functional domains and decreases the chances of interaction among them (*Arai et al., 2001*). It also increases the bioactivity and expression of epitopes (*Bai & Shen, 2006*; *Amet, Lee & Shen, 2009*).

β-defensins are effector antimicrobial peptides having antimicrobial activity including antiviral, antibacterial and antifungal activity (*Lehrer & Lu, 2012*). They recruits immature dendritic cells and naïve T-cell at the site of an infection (*Allaker, 2008*) and use both Th1 and Th2 dependent pathways to boost the synthesis of antigen-specific immunoglobulins (*Tani et al., 2000*). *Kim et al. (2018)* demonstrated that the use of human β-defensins can initiate both innate and adaptive immune response against the conjugated antigen. Several studies suggest that β-defensins efficiently induce a prolonged humoral as well as cellular immune response against a pathogen (*Allaker, 2008*) which makes them a potent antiviral vaccine adjuvant.

The two multi-epitopes Cov-I-Vac and Cov-II-Vac are 317 and 367 amino acids long respectively with a molecular mass of 31.6 kDa and 37.2 kDa, respectively. Both predicted proteins are basic according to theoretical pI values. The instability index and aliphatic index scores calculated using in silico analysis indicated that the vaccine proteins can prove to be stable and thermostable in a test tube, if tested experimentally. The GRAVY score for our vaccines was positive and suggests that both are hydrophobic, which may necessitate the use of micelles for better interaction of the vaccine protein within the polar environment of the body (*Pandey et al., 2018*). The efficacy of subunit vaccines can be improved by addition of nanoparticles-based antigen delivery system as they have smaller size and an ability to deliver the antigens to antigen presenting cells in lymph nodes. Self-assembled micelles, in this regard, are adjuvant of choice for amphiphilic molecules. Historically, these systems have been used to encapsulate hydrophobic drugs in micelle core as reviewed by Trimaille and Verrier (*Trimaille & Verrier, 2015*).

Cov-I-Vac and Cov-II-Vac were found to be antigenic, immunogenic, non-allergenic and strong MHC binders. These factors suggest that the multi-epitopic vaccines will have a robust immune response without allergic reaction.

Molecular Dynamic Simulation studies are mainly conducted to check the stability of vaccine by simulating it in a cellular environment (*Ikram et al., 2018*). However, these
simulation studies take longer time so fast, dynamic simulation, CABS-flex analyses were conducted in this study that give similar results like MD simulations performed at 10ns. Moreover, the RMSF obtained from CABS-flex analysis is comparable to the RMSF of NMR ensembles (*Jamroz, Kolinski & Kmiecik, 2014*).

The efficacy of a multi-epitope vaccine depends on the population for which the vaccine is prepared. The human CTL and HTL based T cell epitopes included in our multi-epitope vaccines were 99.29% similar to those of the world population, indicating that it holds great potential to be an effective approach to combat the ongoing pandemic. In order to induce effective immunological responses in the body, the strong binding of the vaccine products with immune receptors is necessary. The molecular docking analysis of our designed multi-epitope based vaccine showed significant interaction with an immune receptor, TLR8, further predicting the efficacy of both vaccines. However, the vaccine Cov-I-Vac showed a stronger interaction with TLR8. The docking scores indicate that the epitopes predicted in this study have higher binding affinity with TLR8 and thus, have the ability to elicit an effective immune response. TLR8 recognizes the single stranded RNA and acts as a major switch to elicit protective immune responses. It initiates antiviral responses via MyD88-dependent pathway leading to the production of type I and III interferons and other pro-inflammatory cytokines via NF-κB activation (such as IL-1, TNF-α, IL-6) (*Lester & Li, 2014*; *Prompetchara, Ketloy & Palaga, 2020*). These immune responses are otherwise suppressed in case of novel SARS-CoV-2 infection, therefore, interaction of predicted vaccine with TLR8 would produce a diverse set of immune mediators leading to initiation of a protective immune response and result in viral clearance, ideally (*Felsenstein et al., 2020*).

The CAI value after in silico cloning in vector suggests that translational efficacy, mRNA codons, of both multi-epitope vaccines is compatible with the host system. Moreover, it also indicates that synthesis of both vaccines is experimentally possible with higher expression within the *E. coli* K-12 system. The advantages of using pUC19 plasmid for cloning of Cov-I-Vac and Cov-II-Vac are high copy number, smaller sized plasmid and color differentiation upon over expression (color of recombinant colonies appears white while non-recombinant ones appear blue) (*Yanisch-Perron, Vieira & Messing, 1985*). The in silico cloned multi-epitopes can be used for experimental validation of these results. In silico cloning of vaccines in vectors has also been performed in studies suggested by *Dar et al. (2019)*.

We believe that immunoinformatics approaches are useful to design effective vaccine candidates that when tested in experimental studies yield proposed outcomes (*Ikram et al., 2018*). All the ORFs of SARS-CoV-2 were subjected to epitope mapping; however, it is noteworthy to mention that no epitope from 3′ UTRs had strong binding affinity with HLA super family alleles. Therefore, no epitopes from this region are part of either CovI-Vac and CovII-Vac vaccines. However, the multi-epitopes do contain strong HLA superfamily allele binding regions from structural as well as non-structural proteins. These structural, as well as non-structural proteins, play an important part in viral assembly and attachment and specific immune responses against them are required to combat the infection (*Lorente et al., 2016*; *Astuti, 2020*). Despite the numerous benefits

associated with this computational study, experimental validation is required to verify these results.

## CONCLUSIONS

The study was aimed to predict a vaccine against SARS-CoV-2 using immunoinformatics approaches. Two multi-epitopes were designed, one without adjuvant (Cov-I-Vac) and the other with β-defensin adjuvant (Cov-II-Vac). The in silico designed multi-epitopes Cov-I-Vac and Cov-II-Vac are predicted to be antigenic, immunogenic, thermostable and safe for human applications. The multi-epitopes cover the predicted proteome of SARS-CoV-2 (Fig. 2). Cov-I-Vac demonstrates better interaction with TLR8 and more energy is required to deform the structure as compared to Cov-II-Vac. Future prospects of the study involve experimental validation of these results.

## ACKNOWLEDGEMENTS

We would like to thank administration of Atta ur Rahman School of Applied Biosciences (ASAB) for the administrative support needed to conduct this study. Moreover, we would also like to express our gratitude to Ms Brenda Oppert for her feedback in refining the manuscript. Lastly, we want to thank reviewers for their thoughtful comments and efforts towards improving the manuscript.

### Funding

The authors received no funding for this work.

### Competing Interests

The authors declare that they have no competing interests.

### Author Contributions

- Tahreem Zaheer conceived and designed the experiments, performed the experiments, analyzed the data, prepared figures and/or tables, authored or reviewed drafts of the paper, and approved the final draft.
- Maaz Waseem conceived and designed the experiments, performed the experiments, analyzed the data, prepared figures and/or tables, authored or reviewed drafts of the paper, and approved the final draft.
- Walifa Waqar conceived and designed the experiments, performed the experiments, analyzed the data, authored or reviewed drafts of the paper, and approved the final draft.
- Hamza Arshad Dar conceived and designed the experiments, analyzed the data, authored or reviewed drafts of the paper, and approved the final draft.
- Muhammad Shehroz conceived and designed the experiments, performed the experiments, analyzed the data, prepared figures and/or tables, authored or reviewed drafts of the paper, and approved the final draft.

- Kanwal Naz conceived and designed the experiments, performed the experiments, analyzed the data, prepared figures and/or tables, authored or reviewed drafts of the paper, and approved the final draft.
- Zaara Ishaq performed the experiments, analyzed the data, authored or reviewed drafts of the paper, and approved the final draft.
- Tahir Ahmad conceived and designed the experiments, analyzed the data, authored or reviewed drafts of the paper, and approved the final draft.
- Nimat Ullah analyzed the data, prepared figures and/or tables, authored or reviewed drafts of the paper, and approved the final draft.
- Syeda Marriam Bakhtiar analyzed the data, authored or reviewed drafts of the paper, and approved the final draft.
- Syed Aun Muhammad analyzed the data, authored or reviewed drafts of the paper, and approved the final draft.
- Amjad Ali conceived and designed the experiments, analyzed the data, authored or reviewed drafts of the paper, and approved the final draft.

## Data Availability

The raw measurements are available in the Supplemental Files.

## Supplemental Information

Supplemental information for this article can be found online at http://dx.doi.org/10.7717/peerj.9541#supplemental-information.

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
