# Peer review of "Anti-COVID-19 multi-epitope vaccine designs employing global viral genome sequences"

_PeerJ, doi:10.7717/peerj.9541_

## Round 0.1 · original submission · Major Revisions

Please find attached an edited file with some suggestions to clarify and improve the manuscript. Please make sure that the language indicates that this is an in situ study, so that the reader knows that the vaccines have not been tested. I also did not find table or figure legends, so please ensure these are uploaded with your revised file. Fig. 1 was blank for me, even though it appeared there was content by the file size, so you may want to upload again. Once I have received your revisions, I will send the manuscript for scientific review.

---

## Round 0.2 · Major Revisions

Previously I asked you to address questions/comments prior to sending for review, which you have done. After receiving your revised manuscript, I then asked for peer review, and several reviewers have additional comments for you to address. I ask you to review their comments and revise accordingly.

Reviewer 1 ·

Basic reporting

This is a very comprehensive bioinformatic analysis of SarsCov2. the authors included many different immunobioinformatic tools to analyze the virus genome and were able to propose immunogens to be used for the development of a vaccine. The text is clear, professionally written, with few phrases to be rebuilt. The structure is good, well designed, and all data are shown by the authors. Some Figures, as shown in the pdf attached to this review, should have their definition improved. Tables are ok. The results of the bioinformatic analysis are sufficient to justify the author´s hypotheses. The discussion, however, is insufficient in some points and should be improved, mainly in some of the immunological aspects of the use of the new immunogen (please see the attached pdf file).

Experimental design

This work is original, interesting and is included in the journal´s scope.
The research question is well defined. All the aspects included in the study are pertinent.
The results are well presented. The bioinformatic tools used are valid and the association between them is remarkably interesting.
Methodology is sufficient.

Validity of the findings

The findings are interesting, originals. The authors present their findings in a very well structured way.
All data were supplied. The results of the bioinformatic analysis are well presented.
Discussion is not sufficient in several points (please see the attached pdf file).
Conclusions are well stated. Speculations are identified, and are not redundant or even excessive.

Additional comments

This is a very comprehensive and inclusive study. Several bioinformatic analysis were made, and they are well structured and presented.
In some points it would be better not to mention the Pakistani situation so much. Avoiding such a thing would give to the work a more global aspect.
The authors should be more specific when describing the B cell epitope analysis results and discussion. They even state the importance of humoral response, but they do not present these results or even discuss them sufficiently.
The discussion on TLR8 should also be improved.
The definition of some figures should be improved as well.
Please refer to the attached pdf file for more information.

Annotated reviews are not available for download in order to protect the identity of reviewers who chose to remain anonymous.

·

Basic reporting

Some topics of the article are not clear:
line 65 to review the comma position - Scientists are working tirelessly to find a cure, but as of now the impasse
line 77 to change diseases state to disease state
line 78 to put a comma in: "The genome contains six open reading frames (ORFs), including ..."
line 80 to change "envelop" to "envelops"
line 82 to remove "in" and to put a comma in: "cases, even death ..."
line 86 to change "to treat" to "in treating" and put a comma in: "Currently, ..."
line 125 to change "on predicted protein" to "of predicted protein"
line 147 to put "a" before strong
line 229 to change "about the unique" to "into the unique" and change "aspects about protein" to "aspects of protein"
line 371 to change "water refined models" to "refined water models"
line 391 to put "t" in capslock "Supplementary table 2" to "Supplementary Table 2"
line 395 to change "Eigen value" to "Eigenvalue"
line 403 close the parentheses
line 420 to remove one "of", in this line you have double of
line 439 to change "Our subunit vaccines are therefore projected" to "Therefore, our subunit vaccines are projected"
line 466 to change "silico analysis indicate" to "silico analysis indicated"
line 477 to change "was" to "were"
line 493 to change "is" to "are"
line 503 to change "nonstructural" to "non-structural"
line 503 504 to put a comma in: These structural, as well as nonstructural proteins, play an important

Experimental design

Your methods need a better description in the servers that you used. Please, check them:
line 130 In CTLpred, what is the predict approach? (p.e. Quantitative matrices based; Only ANN Based; Only SVM Based; Consensus Approach; Combined Approach) What are the parameters? (p. e. QM Cutoff &nb sp; ANN Cutoff Score; SVM Cutoff Score). Please, answer this question for all tools used in this section.
line 166 In IFNepitope, what is the approach used? (p.e. Motif based; SVM based; Motif and SVM hybrid)
line 201 In AlgPred, what is the method chosen in the server? (p.e. Mapping of IgE epitopes and PID; MEME/MAST motif; SVM module based on amino acid composition; SVM module based on dipeptide composition; Blast search on allergen representative peptides (ARPs); Hybrid Approach (SVMc+IgE epitope+ARPs BLAST+MAST)). Please, answer this question for all tools used in this section.
line 212 In AntigenPro, what are the parameters used? (p.e. ACCpro: Solvent Accessibility (25%); SSpro: Secondary Structure (3 Class); ABTMpro: Alpha Beta Transmembrane; DISpro: Disorder; CONpro: Contact Number; CMAPpro: Contact Map; 3Dpro: Tertiary Structure; SOLpro: Solubility upon Overexpression; ACCpro20: Solvent Accessibility (20 Class); SSpro8: Secondary Structure (8 Class); DOMpro: Domains; DIpro: Disulfide Bonds; SVMcon: New SVM Contact Map; COBEpro: Continuous B-cell Epitopes; ANTIGENpro: Protein Antigenicity; VIRALpro: Capsid & Tail Proteins)
line 223 What are the parameters used in 3Dpro? Please, answer this question for all tools used in this section. Check this website (http://scratch.proteomics.ics.uci.edu/) in line 224, is it the site of 3DPro?
line 237 What are the configurations in HADDOCK 2.2? (p.e. in the first: molecule Active residues, Passive residues, Segment ID to use during the docking and What kind of molecule are you docking?); and in the second molecule: Active residues, Passive residues, Segment ID to use during the docking and What kind of molecule are you docking? Please, describe all parameters used in the server ...)
line 251 Do you use Additional Options in server JCAT?

Validity of the findings

no comment

Additional comments

Your article has an interesting topic in our pandemic time and these methods are very important to find a solution to this problem, but I need to know more details about your methods. I have some doubts.

Reviewer 3 ·

Basic reporting

This manuscript is about immunoinformatics approach of COVID-19. It describes the in silico approaches for the development of multi epitopes based vaccine against COVID-19. The work presented is of standard quality comprising bioinformatics research with all the important data and subsequent analyses included.

Experimental design

The approach is straightforward, however, there are few comments, observations need to be addressed and/or elaborated.

Validity of the findings

The result has novelty and may aid in the development of an effective vaccine against the target virus in future

Additional comments

Comments for the author
This research has significance in the current scenario of the field, but I would recommend some major revision as I found a few major and minor issues throughout the manuscript.
Comments for the author
This research has significance in the current scenario of the field, but I would recommend some major revision as I found few major and minor issues throughout the manuscript.

Major


• In order to global vaccine, a population coverage analysis would generate more acceptable epitope candidates with the ability to confer immunity in different endemic/pandemic regions of the world.
• Authors should mention the references for the linkers and why those linkers are appropriate and were used for vaccine construct.
• There is no mention of EAAAK linker in the manuscript, just in Figure 2.


• In the discussion section mention few similar approaches that have been utilized for in silico vaccine design against other pathogens. Some recent publications are worth citing:
• As described in link provided "Purpose: The aim of this server is to predict MHC Class-II binding regions in an antigen sequence, using quantitative matrices derived from published literature by Sturniolo et. al., 1999. The server will assist in locating promiscuous binding regions that are useful in selecting vaccine candidates". CTL epitopes bind with MHCI molecules, not MHCII.
• Line 140-141: “strong binding affinity” Please specifies the cut-off applied here.

• Line 148-149: “MHC binding …” Please specify the difference on prediction methods among Net MHC 4.0 server and IEDB server as both predict binding affinity with HLA super family alleles.
• Authors should give a in details about the comaprative immunogenic properties the two vaccine construct.

In Table 1 (CTL, mapped epitopes into 6 ORFs - 1AB, S, M, 6A, 7A, and 7B), and in Table 2 ( HTL, they mapped epitopes into 4 ORFs( envelope, 8, 9 and 3a) - covering 10 ORFs, not 11. In addition, it should be clear if they aimed this distribution (because the ORFs are not repeating in each table). If yes, would be good to explain if author selected some epitopes instead of others for this reason.

• In the result section, (Table 1) there are 23 epitopes described as the short-listed CTL epitopes. However, in figure 2 there are only 12 epitopes. Confusing please clarify.

• I would suggest author to give a proper workflow of his analysis with number of epitopes used and discarded.

• Same for (Table 2) HTL epitopes. It is written 11 epitopes in text, there are 9 in table 2 and 10 in figure 2.


Minor

Abstract:
Line 29-30: The aim of the article was find the diversity, maybe "cover the diversity of SARS-CoV-2 strains".
Line 35-37 : “All Open Reading Frames ORFs) were subjected to Cytotoxic T-lymphocyte (CTL) epitope and Helper T cell lymphocyte (HTL) epitopes using CTLpred and HLApred, respectively”
Kindly re-phrase the sentences
Line 64-64: SARS-Cov/(MERS-CoV) is the abbreviation for the virus not the illness.

Line 100: Line 107, Line 108: Author should check trough out the manuscript to maintain the nCOV-19 or SARS-CoV2
Similar problem is multiepitope, multi-epitope etc. please check.
Spacing and end of the sentences are clear eg.
Line 244: Plasmid UC19 was used for in silico cloning of both multi epitopes. More
Line 356: BspQI-sapI (Fig 9. Should be BspQI-sapI (Fig 9).
Line 373: The overall structure of of 475 genomes of COVID-19 …


Please check thoroughly throughout the manuscript.


Line 158: IFN … Which type of IFN?Please specify.
Line 227 : Molecular Docking with Immunological Receptors it should be “Molecular Docking with TLR8 ”as authors is not using other TLRs .
Line 250: “The Pakistan ..” clad should be clade.
Line 258: Please standardize nomenclature of Pakistani strain. “CDS of MT240479” Here its different from Methodology section "genome from Pakistan Accession number GWHACDD01000001"
Line 265-267: “For. Potentially immunogenic “Sentence too long and confusing. re-phrase the sentences

---

## Round 0.3 · accepted · Accept

Thank you for your efforts to address reviewer comments and improve the manuscript.